# The Dangerous Side of Being a Predator: *Toxoplasma gondii* and *Neospora caninum* in Birds of Prey

**DOI:** 10.3390/pathogens12020271

**Published:** 2023-02-07

**Authors:** Stefania Zanet, Fabrizia Veronesi, Giuseppe Giglia, Carolina Raquel Pinto Baptista, Giulia Morganti, Maria Teresa Mandara, Rachele Vada, Luis Manuel Madeira De Carvalho, Ezio Ferroglio

**Affiliations:** 1Department of Veterinary Science, University of Turin, Largo Paolo Braccini 2, 10095 Grugliasco, Italy; 2Department of Veterinary Medicine, University of Perugia, San Costanzo Street 6, 06126 Perugia, Italy; 3Centre for Interdisciplinary Research in Animal Health, Faculdade de Medicina Veterinária, Universidade de Lisboa, Avenida da Universidade Técnica, 1300-477 Lisboa, Portugal

**Keywords:** birds of prey, *Toxoplasma gondii*, *Neospora caninum*, genotyping, Italy

## Abstract

*Toxoplasma gondii* and *Neospora caninum* are apicomplexan protozoa of major concern in livestock and *T. gondii* is also considered one of the major threats and a public health concern. These protozoa have a wide range of intermediate hosts, including birds. The present work aimed to assess the prevalence of these cyst-forming parasites in migratory and sedentary birds of prey. The skeletal muscle and myocardium of 159 birds of prey from Central Italy, belonging to 19 species and recovered across 6 Wildlife Recovery Centers/Care structures along the Italian migratory route, were collected specifically for molecular (PCR) and for histopathological analysis to detect *T. gondii* and *N. caninum*. For the molecular analysis, genomic DNA was extracted. The DNA was tested by sequence typing, targeting GRA6, 529 bp repeated element, B1, PK1, BTUB, SAG2, alt.SAG2, and APICO genes for *T. gondii* and to end-point PCR targeting NC5 gene for *N. caninum*. Thirty-seven out of the one hundred and fifty-nine analyzed samples tested positive for *T. gondii* with a prevalence of 23.27% and nine for *N. caninum*, with a prevalence of 5.66%. Thirty-two sequences were obtained from the thirty-seven isolates of *T. gondii*. Among these, 26 presented alleles compatible with type I strain in 1 or more loci, 4 with type II strain and 2 consisted of atypical strains. *Toxoplasma gondii* genetic variability in birds of prey confirms previous findings of wildlife as reservoirs of atypical strains. Results from the histology showed few protozoal tissue cysts in skeletal muscle (n. 4) and hearts (n. 2).

## 1. Introduction

*Toxoplasma gondii* and *Neospora caninum* are closely related to apicomplexan protozoa with a worldwide distribution. These cyst-forming parasites represent a major concern for livestock, as an important cause of abortions and stillbirths, resulting in production losses [1]. *Toxoplasma gondii* is also of great public health concern, because of its zoonotic potential, resulting in congenital toxoplasmosis and other serious clinical presentations in immune-compromised humans [2].

*Toxoplasma gondii* has a wide range of warm-blooded intermediate hosts, including birds [3]. Birds of prey, due to their specific feeding habits, are particularly exposed to horizontal transmission by feeding on infected small mammals and other birds, or through the consumption of sporulated oocysts by contaminated water or food sources [4,5]. However, the degree of risk of acquiring the infections depends on the different species of birds of prey, in fact, strict carnivorous or scavenger species are more exposed than generalist species [6,7]. It has been speculated that both migratory and sedentary species may act as epidemiological sentinels for environmental contamination at global and local geographical scales, respectively, [8] and, through the migratory flows, may also be informative species for the parasite genetic variability [9,10].

This protozoan shows wide variability in genetic structure with strains classified into three major clonal lineages, i.e., Types I, II, and III, and other additional lineages [11] as well as atypical and recombinant genotypes based on genetic polymorphisms [12,13]. *Toxoplasma gondii* lineages are linked to different virulence and clinical presentations in humans and animal models [14] and also different geographical distribution, with some lineages predominant in specific geographic locations [13]. The importance of each lineage/genotype and its distribution is therefore of utmost importance [15].

In Europe, *T. gondii* strains circulating are known to be Type I, II, and III, with Type II as the predominant in domestic as in synanthropic or wildlife animals. However, a recent study carried out on domestic and wildlife animals in Northwest Italy, showed a higher prevalence of Type I over Type II and Type III, with atypical strains present especially in wildlife [11].

*Neospora caninum* has been reported to infect several bird species in particular pigeons, sparrows, waterfowl, and less frequently, raptors [5,16], but the role of such wildlife animals in the life cycle and in the epidemiological scenario of the parasite has not been yet fully elucidated [5]. Peridomestic and wild rodents as well as lagomorphs, which are among the main prey for several of the species included in the present study, have been demonstrated to play a relevant role in *N. caninum* epidemiology [17,18].

Information on the genetic structure of *N. caninum* is limited and fewer molecular epidemiological surveys have been carried out compared to *T. gondii* [19,20]. Studies conducted mainly on domestic animals (e.g., cattle) from Europe and Central-South America seem to suggest a clonal structure within *N. caninum* populations with clonal sub-populations segregated in different geographical areas [21,22] and with a generally lower rate of genetic exchange between *N. caninum* isolates than in *T. gondii* [20,22]. There is evidence that *N. caninum* isolates differ in virulence in the field [23] but whether its different pathogenicity might be influenced by the genetic diversity within *N. caninum* is currently not defined.

Information on circulating of *T. gondii* and *N. caninum* across wild bird populations in Italy are scarce [24,25,26] and to the authors’ knowledge, only one study was carried out in northern regions to genotype the strains of *T. gondii* from raptors [7]. Therefore, the present study aimed to assess the prevalence of *T. gondii* e *N. caninum* infections in birds of prey in Central Italy, and to expand the available information on the population structure and molecular epidemiology of *T. gondii*.

## 2. Materials and Methods

### 2.1. Animal Sampling and Histology

From November 2017 to October 2020 one hundred and fifty-nine carcasses of wild birds of prey [27] were submitted to the Pathology Service of the Department of Veterinary Medicine, University of Perugia (Italy) for necropsy. All the carcasses belonged to animals that died spontaneously or were humanely euthanized for clinical conditions compromising animal welfare (e.g., gunshot lesions, head trauma, and fractures). The animals came from six Wildlife Rescue Centers or Care Structures located in Umbria, Latium, and Tuscany regions (Central Italy) (i.e., A-C.R.A.S-Lago di Vico (Viterbo, Latium), B-University Teaching Veterinary Hospital (Perugia, Umbria), C-C.R.U.M.A-LIPU (Livorno, Tuscany), D-C.R.A.S-Wild Umbria (Perugia, Umbria), E-C.R.A.S-Mugello (Florence, Tuscany), F- C.R.A.S-Formichella (Orvieto, Umbria)) that were included in a five-year integrated National Plan for Prevention, Surveillance and Response to Arbovirus (i.e., West Nile Virus (WNV) and USUTU Virus) on wild bird mortality [28].

During the necropsy, cadaver condition was registered, and animals showing no autolysis (Code 0) or minimum (Code 1) or mild (Code 2) autolysis were included in the study [29]. The skeletal muscle (right pectoral muscle, 1 × 1 × 0.4 cm) and myocardium (transverse section including both the ventricles, 0.4 cm thick) of animals with Code 0 and Code 1 were specifically collected for histological investigations. However, 0.2 × 0.2 × 1 cm of skeletal muscles and myocardium of all the animals were collected and stored at −20 °C waiting to be submitted for molecular analysis. The histological examinations were performed on 3 μm sections of formalin-fixed paraffin-embedded (FFPE) tissues, stained with hematoxylin and eosin stain (H&E) and evaluated on an Optic microscope (Olympus^®^ BX53) to assess the presence of lesions and cysts consistent with *T. gondii* or *N. caninum*. Ethical approval was not required for this study as the animals were collected as part of a post-mortem monitoring PNA plan.

### 2.2. Molecular Analysis

The tissue samples of skeletal muscle and myocardium of each animal were pooled together and subjected to genomic DNA extraction using the QIAamp^®^ DNA Mini Kit (Qiagen GmbH, Hilden, Germany).

Samples were first tested in parallel with three *T. gondii*-specific PCR protocols targeting GRA6, 529 bp repeated element, and B1 genes [30,31,32]. Positive samples were subsequently tested for multilocus PCR-RFLP (polymerase chain reaction–restriction fragment length polymorphism) sequence analysis on five target genes (i.e., GRA6, PK1, BTUB, SAG2, alt.SAG2, and APICO) for *T. gondii* genotyping [31]. The amplification was carried out using Mini Thermal Cycler (Bio-Rad Laboratories Inc., Hercules, USA). Genetic loci were pre-amplified by multiplex PCR with external primers at the following conditions: 95 °C for 5 min, 30 cycles of 94 °C for 30 s, 55/59 °C for 1 min, 72 °C for 2 min and one ending cycle of 72 °C for 10 min. Then, each individual locus was amplified by nested PCR using internal primers with thermic conditions consisting of: 95 °C for 5 min, followed by 35 cycles of 94 °C for 30 s, 60 °C for 1 min, 72 °C for 1.5 min and one ending cycle of 72 °C for 10 min Type 1 strain RH, type 2 strain ME49 and type 3 strain VEG were used as positive control.

All samples were also tested by *end-point* PCR targeting NC5 gene for *N. caninum* [33]. Non-template and positive controls were included in each run. Positive controls consisted of genomic DNA of *T. gondii* and *N. caninum*, previously amplified and identified by sequencing. Bands of expected size were excised and purified from agarose gel and sent for bidirectional sequencing to a commercial service (Macrogen Europe, Milan, Genome Center, Italy). The obtained sequences were analyzed with BioEdit software (Ibis Bioscience, CA, USA) and compared to those available in GenBank using BLAST software (Basic Local Alignment Search Tool, http://blast.ncbi.nlm.nih.gov/Blast.cgi, accessed on 16 November 2022). NEBcutter software (New England Biolabs Inc.; http://nc2.neb.com/NEBcutter2, accessed on 16 November 2022) was used for virtual RFLP [11].

### 2.3. Statistical Analysis

The association between infection and individual variables including taxonomic Family (Strigidae, Accipitridae, Tytonidae, Falconidae), diet (risky diet—small mammals, birds or scavengers; non-risky diet—reptiles), migratory behavior (sedentary, migratory), Wildlife Rescue Center and age (young, adult, not defined (ND)) was assessed by chi-square test or Fisher’s exact test if the number of observations per category was lower than 5. The Shannon diversity index was used to evaluate the variability among wildlife centers in terms of hospitalized species and their relative abundance [34,35]. This index returns an estimation of the evenness of species in a community (in this case, the community of birds recovered at a center): the higher its value, the higher the diversity of species. The index considers both the number of species and the relative abundance of individuals per species:$$\frac{-\Sigma p_{i}*ln\left(p_{i}\right)}{ln\left(S\right)}$$

With *p_i_* being the proportion of the entire community made up of species *i* and *S* the total number of unique species. Shannon diversity indexes were compared in pairs with the Hutcheson *t*-test to point out any statistically significant (*p*-value < 0.05) difference among centers [36].

With the purpose of comparing the sensitivity of target genes for the detection of *T. gondii*, three different markers were tested (GRA6, B1, and 529 bp repeated element) in a pairwise McNemar test [37]. The significance level was set at *p*-values lower than 0.05. The statistical analyses were performed by using R software (version 4.1.0, 2021).

## 3. Results

Of the 159 wild birds examined, 142 animals were adults, 6 juveniles, and 11 of undefined age and belonging to 19 species (Table 1).

Overall, 37 out of the 159 analyzed samples tested PCR positive for *T. gondii* with a prevalence of 23.27% (CI 95%, 17.38–30.48%), all the specimens showed to be positive at GRA6 target, 29 (18.24%, CI 95% 12.57–25.13%) at 529 bp repeated element and 18 (11.32%, CI 95% 6.85–17.30%) at B1 gene (Table 1). *Toxoplasma gondii* was detected in three of the four taxonomic families of the studied birds of prey. There were sixteen Strigidae (23.88%, CI 95% 18.29–39.58%), fourteen Accipitridae (22.95%, CI 95% 14.19–34.91%) and seven Falconidae (23.33%, CI 95% 11.79–30.60%). Nine samples tested positive for *N. caninum*, with a prevalence of 5.66% (CI 95%, 3.10–11.62%); the positive specimens belonged to four Strigidae (6.15%, CI 2.42–14.78%) and five Accipitridae (8.20%, CI 95% 3.55–17.79%) (Table 1). A co-infection was detected in four individuals, with an overall prevalence of 2.52% (CI 95% 0.98–6.29%), belonging to three different species (tawny owl, long-eared owl, and common buzzard) considered to have a risky diet, since they feed primarily on small mammals and/or birds, or are scavengers.

Ninety animals were assigned to Code 0 or Code 1 based on the preservation status and examined by histology. For both tissues (i.e., skeletal muscle and myocardium) major lesional patterns including myocyte atrophy, degeneration and necrosis, inflammation, fibrosis, and mineralization were occasionally identified. Only rarely, multiple, 40–70 X 30 µm, round to oval protozoal cysts with a thin (0.1 µm) and smooth eosinophilic wall encircling numerous 1–2 mm, crescent-shaped bradyzoites were identified in the sampled tissues of four animals (i.e., four in skeletal muscle and two also in the heart). In a few (n. 2) cases, the evidence of cyst rupture was noticed, in association with the presence of a minimal to mild lympho-histiocytic inflammation. Based on the features and thickness of the walls of the cysts recovered [38] and the molecular positivity to *T. gondii* at both the GRA6, 529 bp repeated element genetic targets, the cyst identities were more consistent with *T. gondii* than *N. caninum*.

Results regarding histopathology are summarized in Table 2.

Statistically significant associations (*p*-value < 0.05) were observed between positivity for *T. gondii* and wildlife center from provenience and the age group. Specifically, the prevalence in the A center was statistically significant (*p*-value < 0.05) higher than in the other centers. Shannon index was highest for the F center (0.77), followed by A (0.65), B (0.64), C (0.56), and E (0.44), meaning that the center F had the highest diversity of species being recovered, while E center had the lowest. The Hutcheson *t*-test was significative (*p*-value < 0.05) for the E center with A, B, C, and F centers; and for the F center with A and C centers, remarking the center E has a significantly (*p*-value < 0.05) lower diversity of species compared to all the others, while center F has a significantly (*p*-value < 0.05) higher diversity of species compared to the A, C, and E centers. Only one species came from the D center. However, no correlation was found between the Shannon diversity index and the prevalence of *T. gondii* in the different centers. For all the other considered variables, no significant association was found (Table 3).

Thirty-two sequences were obtained from thirty-seven isolates of *T. gondii*. Among these, 26 showed in one or more loci alleles compatible with type I strain (22 in sedentary birds and 4 in migrants), 4 with type II strain (3 in sedentary birds and 1 in migrants) and 2 consisted of atypical strains (both found in species with migratory behavior) (Table 4). Missing information for most loci did not allow us to draw sound conclusions on the genotype [38,39].

It was possible to identify a significant difference among the markers used in the detection of *T. gondii* (significance *p*-value: GRA6 and 529 bp repeated element adjusted *p* = 0.08; GRA6 and B1 *p* = 0.0001; 529 bp repeated element and B1 *p* = 0.015). A significantly higher prevalence was obtained using the GRA6 target gene (23.27%, CI 95% 17.38–30.42) (*p*-value < 0.05) compared to B1 (11.32%, CI 95% 7.28–17.18) and 529 bp repeated element (17.61%, CI 95% 12.47–24.27). B1 and 529 bp protocols showed comparable prevalence values (*p* > 0.05).

## 4. Discussion

*Toxoplasma gondii* is a zoonotic parasite for which birds play an important epidemiological role in maintenance and transmission [4]; also *N. caninum* has been shown to infect a wide range of birds, although higher species-specific susceptibility seems to be observed in passerines and pigeons compared to birds of prey [5].

Both parasites have an indirect predator–prey life cycle. Herbivorous, ground-feeding birds can be used as valuable sentinels for assessing the circulation of these parasites in the environment. The prevalence of infection in seagulls and scavenging birds reflects the presence of oocysts in run-off and marine/lake waters, while the prevalence of infection in carnivorous birds reflects the prevalence of infection in their preys [4] with a difference, in the specific case of *T. gondii*, between species that feed primarily on rodents and those feeding on lower-risk preys [6]. The prevalence of infection in carnivorous birds recorded from this study falls within the range previously reported in similar biogeographical areas in Italy [7,25]. Compared to studies carried out in other countries of Western Europe, the prevalence reported in the present study approximates the lower rate observed in France (range 14.3–33%) and Spain (17% to 51%) [4]. The difference among rates of positivity may be related to different prevalence in micromammals in the study areas. Albeit the overall prevalence of infection for *T. gondii* is comparable to these previous reports, the circulating strains reported in the current study seem to differ significantly.

PCR-RFLP based on 12 molecular markers as described by Su et al. [31] represents the current gold standard for genetic characterization of *T. gondii* strains, however direct amplification of the genetic markers from tissues is hampered by the low sensitivity of the method [11], which limits the possibility to obtain useful amplicons for all genetic markers. In Africa and Europe, the two most widespread genotypes are ToxoDB genotypes #1 and #3 (collectively known as Type II), and genotype #2 (known as Type III) [4]. Our results showed a higher circulation of alleles compatible with type I for most loci, which seems to contrast with the high presence of type II reported in birds of prey from different areas of Italy [7]. In the present study, being the genotyping is not complete, the final prevalence of a genotype may differ from the prevalence of single alleles in different loci. However, the frequency and distribution of the alleles here detected are in line with the strains that had been previously reported for several species of wild mammals and livestock from Northern Italy [11].

The statistical analyses showed an association between the positivity for *T. gondii* and the Wildlife Center of origin (*p*-value < 0.05), with a highest prevalence in the A Wildlife Rescue Center (57.14%) when compared to the other centers/care structures enrolled in the study. This association is not related to underlying differences in species hospitalized in each wildlife center nor in their relative abundance as specified by the Shannon diversity index. Further studies should be aimed at clarifying the underlined causes of different prevalence found between the wildlife rescue centers which are located in neighboring but ecologically different areas. Further analysis should include detailed information on the origin of rescued animals and environmental characteristics (i.e., land cover and land use, livestock abundance and density, urbanization level). Age was also identified as a risk factor for *T. gondii* infection. As previously demonstrated, cumulative exposure of birds to the parasite leads to higher prevalence of infection [26,40].

Clinical toxoplasmosis has been sporadically reported in birds of prey [41,42]. Despite this, the association between *T. gondii* seropositivity and the presence of clinical anomalies in birds of prey has been weakly documented [43]. *Toxoplasma gondii* is considered a behavior-altering parasite having deep effects on the hosts [44] and has been shown to modify the behavior and certain physical abilities (i.e., flight and orientation) of several species of birds of prey [45]; moreover, it may directly shape community structure by influencing trophic interactions, food webs, competition, and biodiversity with potential effects on birds of prey.

Since no ante mortem clinical data are available for any of the animals included in the study, the prevalence of infection found in common kestrels *F. tinninculus* is higher than that reported in healthy kestrels from the same country [26]. A possible association between *T. gondii* infection and causes of mortality or hospitalization in wildlife centers (i.e., collisions with cars or buildings) should be further investigated and studies on healthy individuals should be promoted in conjunction with monitoring and ringing activities of free-ranging healthy wild birds.

Several wild avian species have been demonstrated to harbor *N. caninum* DNA [25,46,47,48] as well as tissue cysts [49]. The role of birds in the epidemiology of *N. caninum* has not been completely elucidated. In the present study, *N. caninum* infection was reported in a lower number of species (n. 5) compared to *T. gondii* (n. 10) (Table 1) and in three of these (namely tawny owl, long-eared owl, and common buzzard) both parasites were found to coinfect the same hosts. To our knowledge, this is the first time that DNA of *N. caninum* is detected in tawny owls and long-eared owls, adding *Strix aluco* and *Asio otus* to the list of potential intermediate hosts of the Apicomplexan parasite while confirming the apparent sensitivity of *Buteo buteo* to infection as previously reported [47]. None of the considered variables were significantly associated with infection. Further studies in this regard should be aimed at assessing the genetic diversity of *N. caninum* found in birds of prey and in sympatric wild rodents and lagomorphs [17,18].

Histology analyses are neither sensitive nor specific to detect both the parasites involved in the study, in fact, they show similar tissue cysts morphology as well as cross-reaction on immunohistochemistry [50]. In the present study, only a limited number of animals (n. 4) showed tissue cysts with morphology compatible with *T. gondii/N. caninum*; however, the molecular positivity for *T. gondii* and the traditional lower ability that *N. caninum* has to develop muscle cysts compared to *T. gondii* [51] would allow us to address the cyst identity to *T. gondii*. Regarding the patterns of tissue damage, these were uncommonly identified and always of minimal to a mild degree suggesting a common minimal pathogenic role of these agents in the examined species.

## 5. Conclusions

The results of the study confirmed the circulation of both *T. gondii* and *N. caninum* in birds of prey from Central Italy. However, the molecular analysis did not allow us to draw sound conclusions about the population structure and molecular epidemiology of *T. gondii*, as it was not possible to complete the genotyping. Considering the prevalence of infection found in birds of prey and the non-univocal genetic patterns found in birds of prey, further research is necessary to understand the role of predatory birds and their prey in the maintenance and spread of both *T. gondii* and *N. caninum*.

## Figures and Tables

**Table 1 pathogens-12-00271-t001:** Prevalence of *Toxoplasma gondii* and *Neospora caninum* in birds of prey collected from 2017 to 2020 in six Wildlife Rescue Centers located in Central Italy.

Family	Scientific Name	*T. gondii* Positive/Examined (Prevalence%)	*N. caninum* Positive/Examined (Prevalence%)	Co-Infection with *T. gondii* and *N. caninum* Positive/Examined (Prevalence%)
Strigidae	*Stryx aluco*	5/14 (35.71 %)	1/14 (7.14%)	1/14 (7.14%)
	*Athene noctua*	9/40 (22.50%)	1/40 (2.56%)	0/40 (0%)
	*Otus scops*	1/2 (50%)	0/2 (0%)	0/2 (0%)
	*Asio otus*	1/10 (10%)	2/10 (20%)	1/10 (10%)
	*Bubo bubo*	0/1 (0%)	0/1 (0%)	0/1 (0%)
Accipitridae	*Accipiter gentilis*	0/3 (0%)	1/3 (33,33%)	0/3 (0%)
	*Buteo buteo*	10/26 (38.46%)	4/26 (15.38%)	2/26 (7.69%)
	*Pernis apivorus*	1/7 (14.29%)	0/7 (0%)	0/7 (0%)
	*Accipiter nisus*	3/18 (16.67%)	0/18 (0%)	0/18 (0%)
	*Gyps indicus*	0/1 (0%)	0/1 (0%)	0/1 (0%)
	*Circaetus gallicus*	0/2 (0%)	0/2 (0%)	0/2 (0%)
	*Circus aeruginosus*	0/1 (0%)	0/1 (0%)	0/1 (0%)
	*Aquila chrysaetos*	0/2 (0%)	0/2 (0%)	0/2 (0%)
	*Buteo lagopus*	0/1 (0%)	0/1 (0%)	0/1 (0%)
Falconidae	*Falco biarmicus*	1/1 (100%)	0/1 (0%)	0/1 (0%)
	*Falco peregrinus*	1/8 (12.50%)	0/8 (0%)	0/8 (0%)
	*Falco tinniculus*	5/20 (25%)	0/20 (0%)	0/20 (0%)
	*Falco subbuteo*	0/1 (0%)	0/1 (0%)	0/1 (0%)
Tytonidae	*Tyto alba*	0/3 (0%)	0/3 (0%)	0/3 (0%)

**Table 2 pathogens-12-00271-t002:** Histological results on skeletal muscle and myocardium of birds of prey collected from 2017 to 2020 in six Wildlife Rescue Centers/Structures located in Central Italy.

Lesions	Skeletal Muscle	Myocardium
Myocyte atrophy	12/90	0/90
Myocyte degeneration/necrosis	15/90	0/90
Mineralization	2/90	0/90
Inflammation	11/90	7/90
Fibrosis	5/90	1/90
Protozoal cysts	4/90	2/90

**Table 3 pathogens-12-00271-t003:** Statistical analyses of PCR results of *Toxoplasma gondii* and *Neospora caninum* according with the variables known, considering each independent variable (i.e., taxonomic family, diet, migratory behavior, wildlife rescue center/structure, age).

Variable	Categories	PCR *T. gondii:*		PCR *N. caninum:*	
		Positive/Examined (Prevalence%)	Chi-Square *p*-Value	Positive/Examined (Prevalence%)	Chi-Square *p*-Value
Taxonomic Family	Strigidae	16/67 (23.88%)	0.9863	4/65 (6.15%)	0.4058
	Accipitridae	14/61 (22.95%)		5/61 (8.20%)	
	Falconidae	7/29 (23.33%)		0/29 (0%)	
	Tytonidae	0/3 (0%)		0/3 (0%)	
Diet	Risky	37/158 (23.42%)	1	9/158 (5.7%)	1
	Non-Risky	0/1 (0%)		0/1 (0%)	
Migratory Behavior	Sedentary	28/116 (24.79%)	0.4502	9/116 (7.69%)	0.1135
	Migratory	9/43 (19.95%)		0/43 (0%)	
Wildlife Center/Structure	E	16/28 (57.14%)	0.0005048	2/28 (7.14%)	0.737
	C	4/17 (23.53%)		0/17 (0%)	
	F	4/18 (22.22%)		2/18 (11.11%)	
	D	0/2 (0%)		0/2 (0%)	
	A	1/9 (11.11%)		0/9 (0%)	
	B	12/85 (14.12%)		5/85 (5.88%)	
Age	Adult	30/142 (21.13%)	0.03868	8/142 (5.63%)	0.389
	Young	4/6 (66.67%)		0/6 (0%)	
	ND	3/11 (27.27%)		0/11 (0%)	

**Table 4 pathogens-12-00271-t004:** Analysis obtained by Multilocus sequence typing. The first two columns present the ID of samples and the correspondent species. Remaining columns show the results of virtual RFLP (restriction fragment length polymorphism) for each marker (Na = marker could not be successfully amplified/sequenced, I = allele I, II = allele II).

Sample Number	Scientific Name	Common Name	GRA6	SAG2	alt.SAG2	Apico	PK1	BTUB
29	*Athene noctua*	Little owl	I	I	Na	Na	Na	Na
30	*Strix aluco*	Tawny owl	I	I	I	Na	Na	Na
42	*Athene noctua*	Little owl	I	I	Na	Na	Na	Na
44	*Strix aluco*	Tawny owl	III	I	II	Na	Na	Na
183	*Pernis apivorus*	European honey buzzard	Na	I	Na	Na	Na	Na
198	*Athene noctua*	Little owl	Na	I	I	Na	Na	Na
228	*Strix aluco*	Tawny owl	I	Na	I	Na	Na	Na
230	*Athene noctua*	Little owl	I	Na	Na	Na	Na	Na
232	*Athene noctua*	Little owl	I	Na	Na	Na	Na	Na
248	*Buteo buteo*	Common buzzard	I	Na	Na	Na	Na	Na
262	*Falco peregrinus*	Peregrine falcon	Na	Na	II	Na	Na	Na
264	*Falco tinnunculus*	Common kestrel	I	I	Na	Atypical *	Na	Na
266	*Otus scops*	Eurasian scops owl	I	Na	Na	Na	Na	Na
267	*Strix aluco*	Tawny owl	Na	I	Na	Na	Na	Na
270	*Falco tinnunculus*	Common kestrel	I	Na	Na	Na	Na	Na
273	*Buteo buteo*	Common buzzard	I	Na	Na	Na	Na	Na
275	*Strix aluco*	Tawny owl	Na	Na	Na	Na	Na	Na
276	*Athene noctua*	Little owl	Na	I	I	Na	Na	Na
277	*Falco tinnunculus*	Common kestrel	Na	Na	Na	Na	Na	Na
278	*Falco tinnunculus*	Common kestrel	I	Na	I	Na	Na	Na
279	*Buteo buteo*	Common buzzard	I	Na	Na	Na	Na	Na
280	*Buteo buteo*	Common buzzard	I	Na	Na	Na	Na	Na
281	*Buteo buteo*	Common buzzard	I	Na	Na	Na	Na	Na
282	*Buteo buteo*	Common buzzard	I	Na	Na	Na	Na	Na
285	*Falco biarmicus*	Lanner falcon	Na	Na	Na	Na	Na	Na
290	*Buteo buteo*	Common buzzard	I	Na	Na	Na	Na	Na
298	*Accipiter nisus*	Eurasian sparrowhawk	II	Na	Na	Na	Na	Na
329	*Buteo buteo*	Common buzzard	I	I	Na	Na	Na	Na
330	*Buteo buteo*	Common buzzard	I	I	I	Na	Na	Na
346	*Accipiter nisus*	Eurasian sparrowhawk	I	I	I	Na	Na	Na
347	*Buteo buteo*	Common buzzard	II	Na	II	Na	Na	Na
363	*Falco tinnunculus*	Common kestrel	Na	Na	Na	Na	Na	Na
400	*Athene noctua*	Little owl	I	Na	I	Na	Na	Na
403	*Athene noctua*	Little owl	I	Na	I	Na	Na	Na
404	*Asio otus*	Long-eared owl	I	Na	Na	I	Na	Na
405	*Athene noctua*	Little owl	Na	Na	Na	Na	Na	Na
407	*Accipiter nisus*	Eurasian sparrowhawk	II	Na	Na	Na	Na	Na

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
