# Peer review of "The Dangerous Side of Being a Predator: *Toxoplasma gondii* and *Neospora caninum* in Birds of Prey"

_pathogens, 2023, doi:10.3390/pathogens12020271_

Round 1

Reviewer 1 Report

Dear authors,

Greetings for a very well written and thorough job.

I made the following comments, through which, I think, the manuscript can improve (the initial number refers to the line) :

3-4: Despite I found the title very captivating, I discourage keeping as it is. I would give more an epidemiological cut, as it is the main topic of the investigation. Furthermore, despite rare clinical case reports, T. gondii infection seems subclinical in these birds (https://www.jstor.org/stable/20460275#metadata_info_tab_contents) and calling such infection dangerous it seems a little too much, in my opinion.

27: replace “biomolecular” with “molecular”

42: replace “in” with “for”

48-49: I don’t like the “or, still, through”. Consider removing the entire sentence afterwards.

105: Can you specify which part of the myocardium has been used?

110-111: I assume that formalin should be buffered formalin pH 7.4 ? and then you repeated embedded. Modify it, please

112: replace “method” with “stain”.

117: replace “biomolecular” with “molecular”

122: I would rename “529 bp” as “529bp repeated element” or tgREP529 (as reported in Schares G, Globokar Vrhovec M, Tuschy M, Joeres M, Bärwald A, Koudela B, Dubey JP, Maksimov P, Conraths FJ. A real-time quantitative polymerase chain reaction for the specific detection of Hammondia hammondi and its differentiation from Toxoplasma gondii. Parasit Vectors. 2021 Jan 25;14(1):78. doi: 10.1186/s13071-020-04571-8. PMID: 33494796; PMCID: PMC7830817). I prefer this nomenclature as 529bp is also a simple measure of sequence’s length.  Please correct the same element along the manuscript.

165: after “Pairwise Mcnemar test” add reference.

212: replace “tissual” with “protozoal”.

Table 3: replaced “examinated” with “examined”. Plus, I found a little bit confusing the p value which is always on the first row for each parameter evaluated. Could you please find a way to display such statistical value in a better way (i.e. to understand which is the group significantly higher compared to which) ?

Table 4: the last “Accipiter nisus” (number 407) has to go in italic.

243: what indicates the “p<0.5” in brackets? Is there a missing zero? Plus, the “:” following it, is it correct or should it be a “;” ?

263: replace “10” with “12” (the loci evaluated by Su and colleagues)

287: add “weakly” before “documented”. I honestly didn’t know this paper, but if you read more in detail such alleged correlation between clinical signs and T. gondii infection appears very weak (e.g. they didn’t do histopathology at all; they also included fractures in the clinical abnormalities).

288: add “mammalian” before “hosts”. There are no reports that I know saying such a thing in birds

290: are you extremely sure about what this reference says? I search throughout the article, but I couldn’t find such a thing (about T. gondii and birds of prey).

Entire discussion:

I think you should omit more the clinical data and the importance of clinical toxoplasmosis while focus more on the genetic comparison of other papers. There are a few which you could use to compare your data with other countries (Karakavuk M, Aldemir D, Mercier A, Atalay Şahar E, Can H, Murat JB, Döndüren Ö, Can Ş, Özdemir HG, Değirmenci Döşkaya A, Pektaş B, Dardé ML, Gürüz AY, Döşkaya M. Prevalence of toxoplasmosis and genetic characterization of Toxoplasma gondii strains isolated in wild birds of prey and their relation with previously isolated strains from Turkey. PLoS One. 2018 Apr 18;13(4):e0196159. doi: 10.1371/journal.pone.0196159. PMID: 29668747; PMCID: PMC5906005). Finally, I would mention in the discussion (you just very briefly mentioned) that one important  bias of the present study (as any of these kind) was the lack of amplification of all loci. As an example, having a skim through the table, you can easily see that a lot of isolates were assigned to the lineage I based on just one locus. Such defect might have over-estimated the lineage type one reported here and such topic needs to be discussed in the discussion.

Best wishes

Author Response

The answers for the reviewer are in the file below

Reviewer 2 Report

Dear authors,

The manuscript “The dangerous side of being a predator: Toxoplasma gondii and Neospora caninum in birds of prey” assessed the prevalence of these parasites in birds of prey. It is a very interesting study considering these parasites constitute a public health problem and consequently causes economic losses. The title is very interest to the readers and the manuscript is well structured and well written. The only point I have to disagree with is about the genotype of T. gondii and the conclusions about it. Authors did not use an ideal number of genetic markers to successfully genotype the isolates and authors confirm the genotypes based in the amplification of few markers or just one marker.

Genotype of microorganisms consists in an analysis of a set of genetic markers. We can see this kind of analysis for example, considering techniques like RFLP, microsatellites and SNPs. It is impossible to recognize a genotype based in just one molecular marker.

According Fernadéz-Escobar et al., 2022, the use of an insuficient number of molecular markers can be a problem, since a large part of diversity might be missed or even genotipically different.

Above I present some suggestions and comments.

Abstract: review the last 2 phrases since authors should not confirm the types I, II and atypical strains based on the few molecular markers analyzed.

Introduction:

Lines 41-43: reference 1 cite just N. caninum

Lines 65-66: reference 16 is not compatible with the paragraph. They analyzed the principal game species in Spain.

Line 91: did authors access the population structure in this study? Molecular epidemiology could not be achieved considering the number of analyzed molecular markers.

Material and Methods:

Lines 123-124: according to reference 31, authors applied 5’SAG2 proposed by Howe et al. 1997. In this case, 5’SAG2 must be analyzed together with 3’SAG2. It is not possible to conclude a genotype using only 5’SAG2.

Results:

Lines 232-235: reconsider this paragraph and table 4. As mentioned previously, the number of molecular markers analyzed in this study is not sufficient to conclude a genotype. Some studies suggest a minimum number of markers to define a genotype. Fernández-Escobar et al., 2022 suggest a minimum of four markers and Ferreira et al., 2011 suggest a minimum of eight markers. Genotypes determined by the authors as type I and type II can become atypical ones when other markers are analyzed.

Author Response

The answers for the reviewers 2 are in the file below

Reviewer 3 Report

In this manuscript, authors describe the first report of Toxoplasma gondii and Neospora caninum infection and prevalence in nineteen species of birds of prey collected in six Wildlife Recovery Centers along the Italian migratory route in Central Italy. Few data are available about those parasites in the wild bird population; thus, this study is important, especially because it aims at bridging this gap of scientific knowledge, with the support of statistical analysis and molecular characterization of the parasitic DNA.

Although authors could recover in the histological investigation only six protozoal cysts, positivity recorded in molecular analyses assessed a higher prevalence of 23.27% of Toxoplasma gondii and a low prevalence of 5.66% of Neospora caninum.

The findings support previous outcomes; however, data provided in this survey expand knowledge into the scientific field with a particular impact on molecular epidemiology. In fact, coinfections of the two parasites have been reported, the number of bird species investigated is important and the genotypes of Toxoplasma gondii observed in the present study are different from those reported in other surveys.

The manuscript is clearly laid out containing all elements requested in an original paper format, to be edited in a Special Issue. The title is captivating and the abstract fully reflects the contents of the article. The scientific quality of the work is high, adding important information to relevant topic covering many fields. Therefore, topic and findings of the reviewed article adhere to the scope of the Journal.

Results are rather well described and references are accurate and updated. The language is correct unless for few mistaken, probably due to typing error (for instance, line 83, 107, 226, 259, 275 and 290).

However, few minor revisions are suggested in the section "Materials and methods", paragraph "Biomolecular analysis".

The molecular methods used for the screening of samples and in their characterization by sequence typing of positives are not enough described to allow another researcher to reproduce the results. I would suggest a revision of this part, namely “Molecular analyses”, in order to include the missing information of the methods used. For instance, the amount of eluate at the end of DNA extraction, the PCR master mix compositions and reagents concentrations used should be available to reproduce the tests, as well as the release of the software applied for sequence analysis.

Author Response

The answers for the reviewer 3 are in the file below

Reviewer 4 Report

The study aimed to assess the prevalence of Toxoplasma gondii and Neospora caninum infections in migratory and sedentary birds of prey in Italy. Thirty-seven out of the 159 (23.27%) analyzed samples tested positive for T. gondii and 9 (5.66%) for N. caninum. The authors tested and compared three different genes to amplify Toxoplasma gondii DNA, also they aimed to amplify some genetic markers that are used for Toxoplasma genotyping. The study reported important results for Toxoplasma and Neospora frequency in migratory and sedentary birds of prey. In general, the text is well written, clear, and concise. Important adjustments need to made before publishing.

General comments:

It is not possible to confirm the clonal lineage genotypes by only analyzing five molecular markers (GRA6, PK1, BTUB, SAG2, APICO) as performed in this study. It is necessary to perform the methodology described by Su et al. (2010) and amplify all the 10 molecular markers to obtain the genotype ID.

Did the authors try to amplify the 10 molecular markers (SAG1, SAG2, SAG3, BTUB, GRA-6, C22-8, c29-2, L358, PK1, Apico) preconized for Toxoplasma gondii genotyping (Su et al., 2010) - [31]?

I suggest using the terminology “allele” I or II for the results obtained in the molecular markers analyzed instead of “genotype”. If the alleles are different for any molecular markers, the authors may state the atypical/non-clonal strains. Abstract, results and discussion should be adjusted (e.g., lines 34-36; 232-235; 268-269).

Specific comments:

Line 26-27 – Include the region of the study/ Where the animals were found: Central Italy?

Line 28 – Histology itself does not detect/confirm Toxoplasma or Neospora. Suggestion: “…were collected specifically for biomolecular (PCR) to detect T. gondii and N. caninum and for histopathological analysis.”

Line 29 – Replace for: tissue cysts in skeletal muscle…

Lines 40-42 – Include a reference for Toxoplasma gondii information.

Line 36 – Remove: “and primarily of genotype I”.

Line 50 – “depends on” sounds more suitable.

Line 83 – “…in the field [24], but…”

Line 99 - What was the reference [28] used for? Identify the species of birds? Please, clarify in the text.

Sampling (Lines 99-101):

-        Include the period and the region where the study was conducted.

-        Samples were collected and sent by Wildlife Center: this information should be in the text: how many Centers, the names of Centers (if allowed), and the location.

-        Why were the animals sent to the Pathology Service? Any Surveillance Program purpose?

Line 105 – Why the authors did not collect the brain?

Line 113 – Please, include a reference that was followed to identify and characterize the lesions and cysts consistent with T. gondii or N. caninum. How did the authors confirm, that is, how were the cysts distinguished from other cyst-forming protozoa (eg Sarcocystis)?

Lines 118-120 – What was the contribution of reference [29] for the methodology?

Line 123 – Include the method used: multilocus PCR-RFLP (polymerase chain reaction – restriction fragment length polymorphism)

Correct: “on five target genes: GRA6, PK1, BTUB, SAG2, and APICO”. The markers 5’SAG2 and alt.SAG2 are used to genotype the SAG2 gene.  

Line 131 – Start a new paragraph: “All samples were also tested…”

Line 133 – Which Toxoplasma gondii strains were used for multilocus sequence analysis as positive controls?

Table 1:

Line 189 – The title of Table 1 should be more informative, please include the local and period of the study. Suggestion: Prevalence of Toxoplasma gondii and Neospora caninum in birds of prey from Italy (or even the region – Central Italy?), YEAR (include the period of the study). Do not abbreviate Neospora in the title.

Table 1 – Co-infection (capitalize initial).

Table 1 - As some birds may have more than one common name (depends on the region), please, include the scientific name (species) of each animal.

The number of samples of Strigidae and Tytonidae families should be reviewed. According to the Table 1, there are 67 samples of Strigidae family birds and three samples of Tytonidae, totalizing 161 samples. However, Table 3 presents 65 samples of Strigidae family and one sample of Tytonidae, totalizing 156 samples. In Results section, the authors stated that 159 birds were examined. Please, review the data and adjust the results and tables along the text.

Line 173 – Please, review the percentage result of Strigidae family for Toxoplasma gondii and Neospora caninum according to the correct sampling of Strigidae family birds.  

Line 175-176 – CI 95%?

Line 206 – The title of Table 2 should be more informative. Suggestion: Histological results on skeletal muscle and myocardium of birds of prey from Italy…

Line 226 – Remove “that”

Line 228 – Do not abbreviate the scientific names (Toxoplasma and Neospora) in the title.

Table 3:

- Use the same sequence of Table 1 for sorting the Family: Strigidae, Accipitridae, Falconidae, Tytonidae.

- What criteria was chosen to present the results of the centers? Please, if possible, sort the Wildlife Center from A to F or according to the number of samples (decrescent).

Table 4:

- It is not possible to confirm the genotype by analyzing only 5 markers. Use the term allele instead of genotype: Alleles identification of T. gondii obtained by Multilocus sequence typing.

- It is possible to state that it is an atypical genotype if at least two markers show different alleles. For this reason, the column Final may be removed or the result presented as: inconclusive or atypical.

- The meaning of RFLP should be cited in the title or as footnote.

- Standardize Na (not na) for columns PK1 and BTUB.

Line 250 – Remove the comma.

Line 268 – It is not possible to state: “Our results showed a higher circulation of 268 type I strains”. Please, rewrite.

Line 270 – Review the statement: “The frequency distribution of the three clonal…”. This study did not find the three clonal strains. The sentence needs to be rewritten.

Line 274 – If the authors are allowed to identify the Wildlife Centers, as stated in line 274 (Lazio Wildlife Center), please,

Line 294 – Correct: tinnunculus – “… found in common kestrels (Falco tinnunculus) is…”

Line 297 – Remove comma

Line 300 – Correct: N. caninum DNA

Lines 313-320 – Please, include a brief paragraph regarding histology and PCR results: all the six samples with tissue cysts on histology were positive on PCR? For which protozoa? The samples with histological lesions were positive for protozoa DNA?

Conclusions: The authors need to restate the problem addressed in the paper and do not just suggest further studies. The results confirm the presence of Toxoplasma and Neospora in bird of prey from Central Italy. Include a brief statement regarding the detection of Toxoplasma and Neospora in birds from Wildlife Centers.

“Found in birds of prey” is repetitive (lines 322-323), rewrite the sentence for better comprehension.

Author Response

The answers for the reviewer 4 are in file below

Round 2

Reviewer 2 Report

Dear authors, I appreciate the changes made in the manuscript. There are some studies that genotype T. gondii based in just one or few markers. Although these studies have been published, we cannot take these genotypes into account, since the ideal is that genotyping be performed using 10-11 markers. So, they can be compared to genotypes deposited at Toxo DB.

I suggest authors consider in the abstract and text the following change: example- “Among these, 26 presented alleles compatible with type I strain in one or more loci…”

Please, correct Table 1. Use “scientific name” instead of “common name”

Correct the title of table 4. Remove the world “Genotype”. For example, you can use “Analysis”

I wrote some comments in the attached file.

Author Response

Dear reviewer thank you for the further comments, in the new version all your remarks have been taken into account and the text was modified 
